# Geochemical Footprint of Megacities on River Sediments: A Case Study of the Fourth Most Populous Area in India, Chennai

**Sukkampatti Palanisamy  Saravanan [1,2]**, **Marc Desmet [1,*]**,
**Ambujam Neelakanta Pillai Kanniperumal [2]**, **Saravanan Ramasamy [2]**,
**Nikita Shumskikh [1]** **and Cécile Grosbois [1]**

[1]  E.A. 6293 GéoHydrosystèmes Continentaux, Faculté des Sciences, University of Tours, 37200 Tours, France;
     spsaransp@gmail.com (S.P.S.); nikitashumskikh@gmail.com (N.S.); cecile.grosbois@univ-tours.fr (C.G.)

[2]  Centre for Water Resources, Anna University, Chennai 600025, Tamil Nadu, India;
     nk.ambuj@gmail.com (A.N.P.K.); rsarancwr@gmail.com (S.R.)

**\***  Correspondence: marc.desmet@univ-tours.fr

**Abstract:** An intensive surface sediment survey was carried out over 24 locations from the upstream to downstream sections of two large rivers (Adyar and Cooum) in Chennai (India) during the February dry season of 2015. Trace element concentrations were assessed on a <63 µm fraction using the Geoaccumulation Index ($I_{geo}$) and the newly proposed Geochemical Urban Footprint Index (GUFI), which can be performed to determine the pollution status of any megacity river influenced by urban development. The sediment quality of Chennai's rivers was also compared to worldwide megacity pollution using sediment quality guidelines (SQGs), and a new megacity pollution ranking was determined. The $I_{geo}$ results indicate that the Chennai rivers studied are strongly to extremely polluted regarding trace element content of sediment. Silver (Ag), Cadmium (Cd) and Mercury (Hg) are the most significant tracers of urban contamination. Chromium (Cr) concentrations show an industrial contamination gradient in relation to levels of other trace elements (As, Cu, Ni, Pb, and Zn) at the Chennai megacity scale. The GUFI ranges from moderate to extreme contamination, particularly in the downstream stretches of the two rivers. This spatial trend is related to various point sources and identified at specific sampling stations, with a lack of identifiable buffer zones. According to the worldwide comparison of megacity pollution, Chennai is ranked in fifth position. The present position can be attributed to a number of explanations: a population explosion associated with the unplanned growth of the city and non-controlled point sources of pollution in Chennai's waterways.

**Keywords:** megacities; urban river; sediment; pollution; geochemical index

## 1. Introduction

Rapid urbanization and industrial development over the last four decades have resulted in some serious concerns about water bodies, particularly in megacities [1–6]. For a long time, urban rivers have been associated with pollution because of the practice of discharging untreated domestic and industrial waste into rivers [7–10]. Trace elements and organic compounds present in waters and sediments of urban rivers are one of the major quality issues in many fast-developing cities [11–16]. The presence of certain trace elements may highlight certain sources, for example, trace elements such as Ag, Cd, Cr, Cu, Hg, Ni, Pb and Zn from domestic and industrial effluents [17–20].

Elevated content of trace elements have been reported in river sediments of megacities. In Paris (Seine river basin, France), trace element concentrations in sediment varied by one order of magnitude

in peri-urban streams [21,22]. Changes in trace element concentrations over time appear to be the result of increases in both population and industrial activities [23–25]. In Dhaka (Buriganga river, Bangladesh), a high daily amount of untreated industrial waste is discharged into open water bodies and their adjacent lands [26–28]. Besides, a significant amount of heavy metals in suspended particles is transported from neighboring countries, such as India, by the Teesta and Brahmaputra rivers [2,29].

In India, indiscriminate industrialization and urban development have significantly affected the quality of surface water resources [30,31], and the pollution of surface water bodies has attracted considerable public attention over the past few decades. High levels of trace elements can be observed in sediments of many urban rivers in India, such as the Ganges, Yamuna, Hindon, Narmada, Mithi, and Kanini rivers, etc. [29,32–37], although most of the urban populations still depend on river water sources for their day-to-day consumption [38].

The levels of hydrological pollution in the Chennai zone in the south-eastern part of India have increased in recent years through uncontrolled disposal of waste water and pollutants due to human activities [39,40]. Hence, sediment quality is widely used to assess the environmental risk, which has been extended to water resources and the food chain [20,41–43]. River sediments are an important sink but also a source for assessing heavy metal pollution in rivers as they present a great affinity for the solid fraction [36,44–48] and have a long residence time in fluvial systems [49–52]. Heavy metal pollution is thus a serious environmental concern due to the presence of toxic elements such as Ag, Cd, Cr, Cu, Ni, Pb, and Zn, which have become a serious issue in many fast-developing countries [6,42,53–55]. Heavy metals are accumulating in sediments at levels that are several orders of magnitude higher than the natural geochemical background and surrounding order [56–59]. Studying upstream–downstream sediment is therefore an appropriate approach to assess the gradient and sources of contamination.

The aim of the present study is to identify the present pollution status of Chennai megacity, along the two urban rivers, the Adyar and Cooum, and also to identify the sources of trace elements and their transport along the river to the Gulf of Bengal. Based on this aim, the objectives are the following: (i) establish an upstream–downstream characterization of the fluvial system, (ii) estimate the impact of anthropogenic inputs using a new urban tracer specifically designed for Chennai megacity, (iii) identify polluting hotspots by separating out various urban and industrial sources of incoming pollution, and (iv) compare Chennai's level of contamination with worldwide megacity pollution.

## 1.1. General Setting of the Studied Area

The study was carried out in Chennai, the capital of Tamil Nadu (India), located between 12°50′49″ and 13°17′24″ North latitude and 79°59′53″ and 80°20′12″ East longitude on the Coromandel coast in southern India (Figure 1). Chennai is a metropolitan area, with the fourth highest population and the fifth largest metropolitan area in India. At the world scale, Chennai is the 22nd most populous city in Asia and the 40th most populous city in the world [60]. Chennai Metropolitan Area (CMA) covers three districts of Tamil Nadu state and is an area encompassing 1189 km$^2$. Over recent years, Chennai has experienced an incredible growth of the population in and around the city, with a population increase from 7.04 million in 2001 to 8.65 million in 2011 (www.census2011.co.in). The urban population in CMA surpassed 10.26 million in early 2017 [61] and is estimated to reach 11.12 million by 2021 (population projection given by CMA).

Chennai Metropolitan Area has six main waterways crossing the city of which two, the Adyar and Cooum rivers (Figure 1), are natural rivers flowing east towards the Bay of Bengal. These two rivers also convey storm water from the city's sewage drain network, with the latter serving around 45,000 hut-dwelling families who live in several nearby locations [8,39]. The other waterways are man-made, including the Buckingham Canal (Figure 1), which runs from north to south and intersects both rivers. The Buckingham canal is a 796 km-long freshwater navigation canal in the north part of the city and connects most of the natural backwaters along the coast to the port of Chennai.

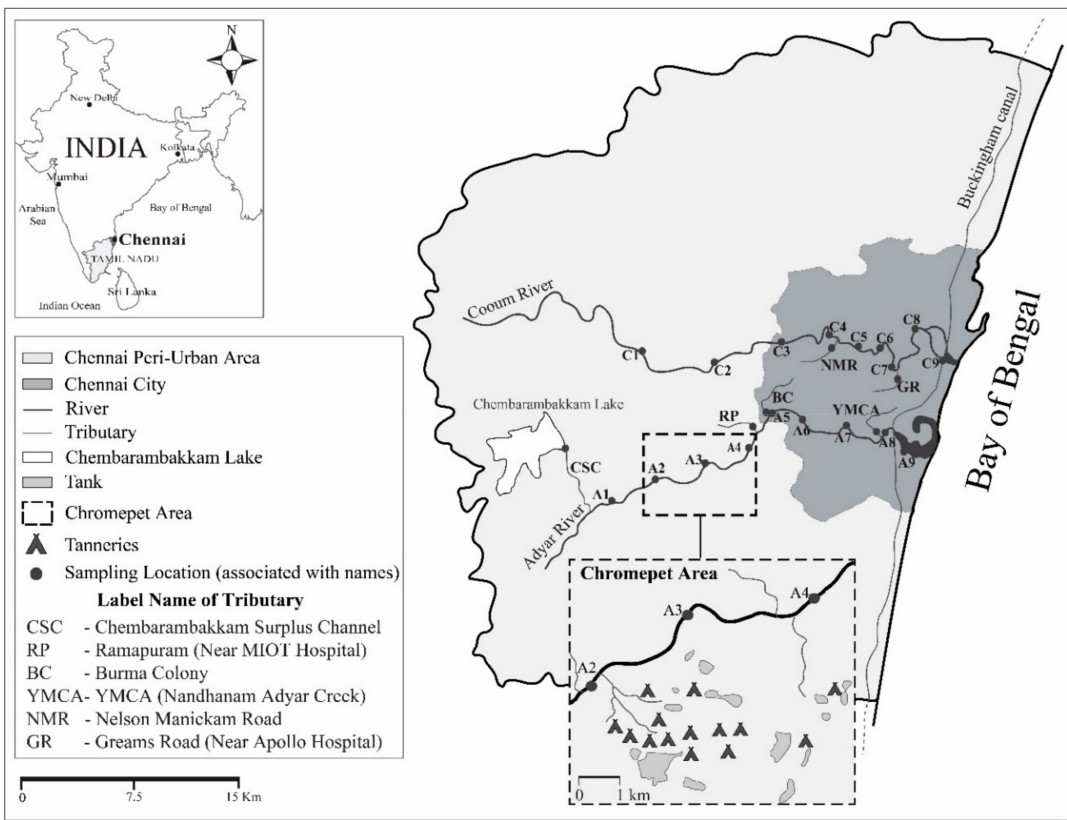

**Figure 1.** Study area showing the sampling sites in Chennai (Tamil Nadu, India).

Chennai lies close to the equator and has a tropical monsoon climate. Maximum temperatures of around 40 °C occur in late May to early June. The minimum annual temperature is 20 °C in early January. The predominant wind direction is from south-east to north-west, and the average annual rainfall is around 1400 mm, falling mainly during the rainy season in October to December.

The economic activities in Chennai establish the largest industrial commercial centre in south India, often referred to as the "Detroit of India" and the "gateway to south India" [62]. In recent decades, large industrial facilities have been established in Chennai suburbs, resulting in large-scale population growth [41]. Chennai City is home to a large base of companies, including petroleum, chemical, rubber, leather, plastic, steel, automobile and manufacturing industries. In addition, multinational companies (MNCs) have set up in the city and include information technology (IT—software services and hardware manufacturing), non-residential complexes (hotels, cinemas and shopping malls), and also medical and health care (Chennai Metropolitan Development Authority, 2008).

### 1.2. Description of the Studied Fluvial System

This study focused on the fluvial system of the Adyar and Cooum rivers and some of their major drains (Figure 1). The drainage basin is a ferricrete surface regolith derived from charnockite bedrock formed due to more or less fragmental decomposed matter drifted by wind, water or other sources of erosion. Almost the entire area is covered by Pleistocene/recent alluvium, deposited by the two rivers. This alluvium is made up of mainly clays, sands, sandy clays and occasional boulder or gravel zones. Sandy areas are found along their banks. Igneous/metamorphic rocks are found in the Adyar river. The river sediments contain clay, silt, and sands originating from the alteration of charnockite rocks and reworked alluvium deposits. A thin layer of laterite is also found in some places, and well-rounded pebbles have been encountered at several locations at varying depths [8,39,63,64].

The length of the Adyar river is 42 km, and the catchment area is 860 km$^2$. Surplus water from the Chembarambakkam reservoir can cause flooding in the river, and a historical flood event was recorded

in Chennai in 2015. The Cooum river is about 65 km long with about 18 km within the Chennai limits. The total catchment area is about 290 km$^2$. Both the Cooum and Adyar are seasonal, and their main flow is due to the north-east monsoon from October to December. Portions of the upstream side of the two rivers will be dried during the non-monsoon period [65], thus they both mostly act as city sewage networks in their respective upstream reach [8]. Sewage and waste water outflows enter the rivers via drains, which are connected to different networks and collect all kinds of urban and industrial waste from the city [65]. River slopes are very low, and sediment transportation is also reduced [66] due to the static condition of river water flow. Thus, the mouths of these rivers open into the Bay of Bengal. The most downstream part of the Adyar experiences a tidal influence, which allows saltwater to enter the mouth [8,35,67].

## 2. Materials and Methods

### 2.1. Sampling and Sample Preparation

Sediment samples were collected from 24 locations (Figure 1) from upstream to downstream sites of both rivers (Adyar river stations: A1–A9; Cooum river stations: C1–C9) and some major drains (corresponding to station names as: A-CSC, A-RP, A-BC, A-YMCA along the Adyar river and C-NMR and C-GR along the Cooum river; Figure 1). Sampling sites were selected according to potential point sources of pollution (industry and urban sewage) and accessibility. The most upstream stations on the Adyar and Cooum rivers were considered to be in the Chennai peri-urban area: stations A1–A4 and C1–C2 (Figure 1). Downstream stations were considered to be in Chennai City (stations A5–A9 and C3–C9). Surface sediments and flood deposits were collected under water at a sediment depth of 0–15 cm using an Uwitec coring device in February 2015. This upper layer of surface sediment is considered as recent deposition from one hydrological cycle/one year and flood deposits. Collected sediment samples were immediately transferred to airtight sealed polythene bags and stored at 4 °C in an icebox during sampling campaigns. They were then transferred to the laboratory and after removing coarse shell fragments, lithoclasts and other organic macro-remains, sediments were dried at 60 °C for 72 hours in a well-ventilated oven. Once dried, samples were desegregated in an agate mortar and sieved through a disposable Nylon mesh to produce a <63 µm fraction. During the drying, grinding, sieving and storage processes, care was taken to avoid contamination.

### 2.2. Analysis of Sediment

Geochemical analyses were performed on the <63 µm fraction in order to limit the grain size effect on trace element concentrations. Analyses were conducted at the national CNRS-SARM laboratory, Nancy, France [68]. Samples were completely digested with $LiBO_2$-$Li_2B4O_7$ in a tunnel oven and placed in an acidic solution. After samples had been dried, the residues were completely re-dissolved with nitric acid ($HNO_{3)}$. Total contents of major and minor elements were analyzed using ICP-AES, and trace elements using ICP-MS (Perkin Elmer 5000) except Hg, for which cold vapor AAS was used (Perkin Elmer 5100). All the digestion processes and analyses were quality-checked through analysis of duplicate samples and internal reference materials [69–72]. Accuracy was within 5% of the certified values and analytical errors less than 10% Relative Standard Deviation for trace element content. Levels detected were at least 30 times higher than detection limits.

Carbon-coated thin sections of sediments were observed using a Zeiss field emission scanning electron microscope with an acceleration voltage of 20 kV, at a working distance of 9 mm and coupled to an energy dispersive X-ray system for element identification and mapping.

### 2.3. Assessment of Sediment Contamination

To assess the sediment contamination, the most common method is to use geochemical indexes such as enrichment factors (EFs) [28,73] and the Geoaccumulation Index ($I_{geo}$) [27,49,62,74–76]. In this study, a new index was calculated, the Geochemical Urban Footprint Index (GUFI), to establish the

pollution status of Chennai megacity's rivers. Unlike EF and $I_{geo}$, this quantitative index is designed to determine a multi-trace element contamination. Finally, to compare the level of Chennai City's contamination to that of worldwide megacities, trace element contamination in sediments of various megacities was compared to numerous sediment quality guidelines (SQGs).

### 2.3.1. Calculation of $I_{geo}$ for This Study

The intensity of heavy metal contamination can be evaluated using the Geoaccumulation Index ($I_{geo}$) as first proposed by [65]. It is mathematically expressed as:

$$I_{geo} \ = \ \log_2 \left\{ \frac{(Me)_{Sample}}{1.5 \times (Me)_{Background\ Sample}} \right\} \tag{1}$$

where $(Me)_{Sample}$ is the measured concentration of an element in the sediment and $(Me)_{Background\ Sample}$ is the geochemical background concentration of the element. In Tamil Nadu, the bedrock consists of charnockites, and its trace element composition has previously been studied [77,78]. However, values of trace elements such as Ag, Cd, and Hg are not available in these articles. Therefore, in the present study, the background values used for $I_{geo}$ calculation correspond to those of standard shale composition described by [79]. A factor of 1.5 as the denominator was used to minimize the effect of possible variations in background values due to lithology [76]. The $I_{geo}$ quantification trace element accumulation in sediments is divided into seven classes (0–6) from unpolluted to extremely polluted [75].

### 2.3.2. Geochemical Urban Footprint Index (GUFI)

This study developed the GUFI based on the concentrations of silver (Ag), cadmium (Cd), and mercury (Hg), some of the most commonly analyzed toxic trace elements in an urban environment [25,28,47]. These three trace elements appeared to be quite sensitive regarding the heavy metal concentrations in the sediment of both rivers. They are influenced by anthropogenic activities such as urban development, industrialization, coal combustion and population density, and they are considered to be sensitive urban tracers based on $I_{geo}$ results for this study. Other toxic trace elements including As, Cr, Cu, Ni, Pb and Zn, when available for studies in an urban context, are ubiquitous as they present various anthropogenic sources.

The GUFI was used to determine the level of contamination of sediments, upstream to downstream of the urban area. First, the Metal Urban Index (MUI) was compared to the most upstream sample in the studied river basin where there was no/limited anthropogenic influence. Ratios to aluminium (Al) concentrations were also calculated as Al was conservative along both the rivers studied. The MUI is a similar pollution index to the enrichment factor. The MUI was calculated as follows:

$$\text{Metal Urban Index (MUI)} \ = \ \left\{ \frac{\left(\frac{Me}{Al}\right)_{Sample}}{\left(\frac{Me}{Al}\right)_{Most\ Upstream\ Sample}} \right\} \tag{2}$$

where $(Me/Al)_{Sample}$ is the ratio of metal (Me) and aluminium (Al) concentrations in the sediment sample and $(Me/Al)_{Most\ Upstream\ Sample}$ is the same ratio for the most upstream sample of the river basin studied.

Once the MUI had been determined for Ag, Cd and Hg, the GUFI was calculated as follows:

$$\text{Geochemical Urban Footprint Index (GUFI)} \ = \ \left\{ \frac{(MUI)_{Ag} \ + \ (MUI)_{Cd} \ + \ (MUI)_{Hg}}{3} \right\} \tag{3}$$

where $(MUI)_{Ag}$, $(MUI)_{Cd}$ and $(MUI)_{Hg}$ are the metal urban indexes for Ag, Cd and Hg, respectively. The GUFI range is classified into six categories of contamination:

- Range 1–10          extremely low contamination
- Range 10–25        low contamination
- Range 25–50        moderate contamination
- Range 50–75        high contamination
- Range 75–100      very high contamination
- Range >100         extremely high contamination

### 2.3.3. Sediment Quality Guidelines (SQGs)

Sediment quality guidelines (SQGs) are scientific tools that synthesize information regarding the relationships between sediment concentrations of metals and organics and any adverse biological effects [47,48,56]. SQGs are defined with a lower and upper effect concentration level [80–84], including a threshold effect concentration (TEC) and a probable effect concentration (PEC) [85–87]. When available in the literature, ubiquitous Cr, Cu, Ni, Pb and Zn are the most frequently analyzed in surface sediments of various megacities. Their concentrations were compared to their respective quality guidelines, and a comparative ranking of various megacities for ubiquitous trace elements was carried out as follows:

$$\text{Mean PEC quotient of a megacity } = \left\{ \frac{\sum_{i=1}^{n} \frac{(\text{Me})_{i \text{ Sample}}}{(\text{Me})_{\text{PEC Sample}}}}{n} \right\} \tag{4}$$

where $(\text{Me})_{\text{Sample}}$ is the measured concentration of metal in the sediment, $(\text{Me})_{\text{PEC Sample}}$ is the probable effect concentration for this Me in the SQGs [87], and n is the number of trace elements taken from a megacity. The mean concentration of a sediment sample is considered for the trace element comparison of megacities. This mean PEC quotient was thus used to determine the megacity ranking, with a higher quotient indicating a more impacted megacity.

## 3. Results and Discussion

### 3.1. Upstream–Downstream Trace Element Gradient in Urban Sediments

Sediment concentrations of trace elements are presented in Table 1 according to kilometric points (KP, in km; KP 0.0 being the most upstream station sampled).

For both rivers, the minimum concentrations of trace elements Ag, Cd, Cu, Hg, Pb and Zn were always found in sediment samples collected at the most upstream stations (A1 and C1, Table 1). Concentrations at these stations were in the same range of magnitude as those in sediments collected from the most upstream drain, station A_CSC at the Chembarambakkam Surplus Channel, in the Chennai peri-urban area. These concentrations indicate that the upstream stations can be considered as not influenced by any anthropogenic activity regarding the concentrations of Ag, Cd, Cu, Hg, Pb and Zn levels. In contrast, the minimum concentrations of As, Cr and Ni were measured in more downstream reaches, especially at the mouths of the rivers. The low concentrations observed at stations A9 and C9 correspond to a physico-chemical desorption of trace elements due to an increase in the salinity of the urban waters. The kinetics of trace elements are highly sensitive to pH, salinity and particle loading.

In the Adyar river, the highest sediment concentrations were found at station A6 for Ag, Cd, and Cu and at station A8, 6 km downstream from A6, for As, Hg, and Pb. These results indicate an increase in these trace elements' concentrations (Ag, As, Cd, Cu, Hg and Pb) from upstream to downstream of the Adyar river. For As, Cr and Ni, the minimum concentrations were found at station A5, and their concentrations' maxima were measured at different stations: A8 for As, A2 for Cr and A5 for Ni.

In the Cooum river, station C6 presented the highest levels of Ag, Cd, Hg and Zn, with station C9 for Cu, indicating an up–downstream gradient of contamination along the river for these trace elements as for the Adyar river. For As, Cr and Ni, the lowest concentrations were measured in the middle of the river length at station C7 and their maxima at station C9, the most downstream station.

For both rivers, all trace element concentrations in sediments decreased at the most downstream station in the estuarine part (stations A9 and C9) except for As, Cr, Ni and Pb in Cooum sediments.

**Table 1.** Trace element and aluminium concentrations (mg/kg) in the Adyar river basin (A stations) and Cooum river basin (C stations) from upstream to downstream, with KP 0.0 being the most upstream station sampled, and their associated drains (see Figure 1 for locations).

| | Station | Kilometric Point (km) | Ag | As | Cd | Cr | Cu | Hg | Ni | Pb | Zn | Al |
|---|---|---|---|---|---|---|---|---|---|---|---|---|
| Adyar River | A1 | 0.0 | 0.1 | 3.1 | 0.2 | 141 | 68 | 0.0 | 66 | 22 | 115 | 93,100 |
| | A2 | 3.0 | 0.8 | 3.1 | 0.8 | 1517 | 237 | 0.1 | 67 | 55 | 1392 | 76,000 |
| | A3 | 6.6 | 0.6 | 2.4 | 0.4 | 925 | 82 | 0.1 | 62 | 28 | 155 | 86,300 |
| | A4 | 12.0 | 0.2 | 1.8 | 0.4 | 700 | 83 | 0.1 | 76 | 24 | 145 | 77,100 |
| | A5 | 14.7 | 1.8 | 1.8 | 0.8 | 74 | 74 | 0.3 | 18 | 26 | 122 | 54,600 |
| | A6 | 15.9 | 18.8 | 3.7 | 4.8 | 348 | 325 | 1.4 | 55 | 54 | 722 | 61,200 |
| | A7 | 18.4 | 7.1 | 2.9 | 1.5 | 164 | 182 | 0.5 | 44 | 60 | 308 | 67,500 |
| | A8 | 21.9 | 12.1 | 4.1 | 2.3 | 359 | 192 | 1.8 | 56 | 65 | 321 | 76,400 |
| | A9 | 23.7 | 4.4 | 3.1 | 1.1 | 167 | 85 | 0.5 | 28 | 33 | 155 | 62,400 |
| | Min. | - | 0.1 | 1.8 | 0.2 | 74 | 68 | 0.0 | 18 | 22 | 115 | 54,500 |
| | Max. | - | 18.8 | 4.1 | 4.8 | 1517 | 325 | 1.8 | 76 | 65 | 1392 | 93,100 |
| Cooum River | C1 | 0.0 | 1.2 | 4.9 | 0.4 | 109 | 61 | 0.1 | 46 | 38 | 148 | 82,900 |
| | C2 | 4.1 | 9.3 | 3.8 | 1.1 | 185 | 188 | 0.1 | 46 | 340 | 381 | 80,400 |
| | C3 | 9.2 | 2.6 | 4.5 | 0.9 | 123 | 86 | 5.7 | 44 | 52 | 203 | 81,300 |
| | C4 | 12.8 | 17.7 | 3.9 | 9.9 | 113 | 179 | 3.7 | 42 | 79 | 517 | 73,800 |
| | C5 | 14.6 | 22.4 | 4.3 | 16.2 | 119 | 254 | 1.7 | 47 | 71 | 586 | 70,400 |
| | C6 | 15.8 | 31.0 | 3.7 | 17.4 | 105 | 223 | 6.6 | 39 | 84 | 595 | 71,900 |
| | C7 | 17.4 | 16.1 | 3.5 | 7.3 | 95 | 155 | 4.3 | 36 | 58 | 382 | 73,500 |
| | C8 | 19.8 | 15.4 | 4.0 | 10.1 | 104 | 173 | 1.5 | 36 | 71 | 471 | 71,800 |
| | C9 | 23.5 | 9.0 | 10.8 | 7.8 | 226 | 302 | 1.8 | 49 | 145 | 402 | 77,300 |
| | Min. | - | 1.2 | 3.5 | 0.4 | 95 | 61 | 0.1 | 36 | 38 | 148 | 70,400 |
| | Max. | - | 31.0 | 10.8 | 17.4 | 226 | 302 | 6.6 | 49 | 340 | 595 | 82,900 |
| Tributary | CSC | - | 0.1 | 2.6 | 0.4 | 96 | 45 | 0.0 | 37 | 17 | 65 | 82,400 |
| | RP | - | 7.2 | 3.9 | 0.7 | 267 | 159 | 1.7 | 61 | 59 | 776 | 74,200 |
| | BC | - | 9.7 | 3.0 | 0.9 | 226 | 336 | 1.8 | 56 | 75 | 457 | 63,800 |
| | YMCA | - | 14.3 | 3.9 | 10.0 | 131 | 239 | 1.4 | 44 | 65 | 415 | 70,200 |
| | NMR | - | 14.5 | 4.3 | 37.6 | 95 | 346 | 1.5 | 40 | 54 | 442 | 64,500 |
| | GR | - | 5.9 | 4.4 | 0.5 | 93 | 104 | 1.0 | 36 | 59 | 358 | 70,700 |
| | Min. | - | 0.1 | 2.6 | 0.4 | 93 | 45 | 0.0 | 36 | 17 | 65 | 63,800 |
| | Max. | - | 14.5 | 4.4 | 37.6 | 267 | 346 | 1.8 | 61 | 75 | 776 | 82,400 |

The specific city of Chennai is related to the presence of aerial water drains, which collect dumped solid waste, unchecked discharge of untreated industrial effluents (mechanical, steel, plastics, fiber, textile, and surgical instrument factories, petrochemical industries and car workshops), effluent from a large number of educational institutions, hospitals and hotels, and domestic sewage from slums spreading extensively along the river banks. A total of 58 drain outlets discharge 0.775 million liters per day (MLD) of industrial effluent and 8.1 MLD of domestic sewage into the Adyar river between stations A6 and A9 [8,39]. Similarly, in Cooum river, 158 sewage/storm water outlets bring raw sewage and untreated water into the city limits. The Cooum river receives 0.4 MLD of industrial effluent and ten times more domestic sewage (80 MLD) from the drains than the Adyar river [66]. Therefore, several drains were sampled in order to quantify trace element levels from some potential punctual pollution sources as they are considered to be representative of specific urban inputs. The drain stations A-RP and C-GR received waters coming from hospital wastes. Drain stations A-BC, A-YMCA and A-NMR are the three main urban drains of Chennai City, collecting waste water from various urban areas. A comparison of trace element levels in drains and in the two studied rivers shows that the drain



maxima were in the same range of concentrations, presenting even more elevated concentrations than river sediments for Cd, Cu, Ni and Zn.

### 3.2. Identification of Suitable Urban Tracers

The most widely adopted pollution index, the Geoaccumulation Index ($I_{geo}$), was calculated in this study; its spatial variations (Table 2) were used to identify specific urban tracers in river sediments of Chennai megacity. Median $I_{geo}$ values were calculated for the upstream peri-urban area, considered for the Adyar river to be between KP 0 and 14 (stations A1–A4), for the Cooum river, from KP 0 to 9 (stations C1 and C2), and for the urban area, downstream of these stations up to the estuary (Table 2).

In the upstream basins of both rivers, all the mean $I_{geo}$ values are lower than 3 for all the trace elements except Cr in the Adyar river ($I_{geo}$ up to 4.8) and Ag in the Cooum river ($I_{geo}$ up to 6.8). For chromium, the upstream part of the Adyar drainage basin is located in the Chromepet area (Figure 1), where there are many industries including metallurgy, electroplating, paint and pigment production, tanneries, and paper pulp production. The tanning industry is a well-known contributor to chromium pollution of water resources [2,88]. Around 152 tanneries are clustered in the Chromepet area. Before the building of a waste water treatment plant in 1995, untreated effluent was discharged into the low-lying areas of the Adyar river [89]. An increase in the chromium content of sediments, also for Ag, Cd, Cu and Ni, in this area would suggest that leaking and/or malfunctioning of the treatment plant is still occurring in the Chromepet district and that meteoric water percolating through the soil still lixiviates the Chromepet residual solid wastes. For silver in the upstream part of the Cooum river (station C2), this elevated $I_{geo}$ is associated with elevated $I_{geo}$ for Cd, Cu, Pb and Zn. They probably correspond to punctual but not yet identified inputs and to atmospheric deposition of finer particles from smelting operations, electrical products and coal combustion.

**Table 2.** Geoaccumulation Index ($I_{geo}$) of trace elements in surface sediments for the Adyar and Cooum rivers.

| $I_{geo}$ Value–Class | Pollution Level of the River Sediments | Mean Value of Chennai Peri-Urban Area | | Mean Value of Chennai Urban Area | |
|---|---|---|---|---|---|
| | | Adyar River (A1–A4) | Cooum River (C1 and C2) | Adyar River (A5–A9) | Cooum River (C3–C9) |
| 0 ($I_{geo} < 0$) | Unpolluted | As, Hg | Hg | - | - |
| 1 ($0 < I_{geo} < 1$) | From Unpolluted to Moderately polluted | Pb | As, Ni | As, Ni, Pb | As, Ni |
| 2 ($1 < I_{geo} < 2$) | Moderately polluted | Ag, Cd, Ni, Zn | Cr, Zn | Cr, Zn | Cr, Pb |
| 3 ($2 < I_{geo} < 3$) | From Moderately polluted to Highly polluted | Cu | Cd, Cu, Pb | Cu | Zn |
| 4 ($3 < I_{geo} < 4$) | Highly polluted | Cr | - | Cd, Hg | Cu |
| 5 ($4 < I_{geo} < 5$) | From Highly polluted to Extremely polluted | - | - | - | - |
| 6 ($I_{geo} > 5$) | Extremely polluted | - | Ag | Ag | Ag, Cd, Hg |

The mean $I_{geo}$ value in Chennai City sediments was below class 2 for As, Cr, Ni and Pb for both rivers and for Zn level in Adyar river sediments only (Table 2). Adyar river sediments are moderately to highly polluted in Cd, Cu and Hg, and Cooum river sediments in Cu and Zn. The most elevated $I_{geo}$ mean value was for Ag in Adyar river and for Ag, Cd and Hg in Cooum river sediments, corresponding to class >5, which indicates a highly to extremely polluted state. Specifically, sources of Ag can be related to metallurgy activities, electronic products, medical equipment and/or practices, health care product use (soap, shampoo, cream, etc.), jewellery, antimicrobial agents used in fibers and water purification systems [45]. Some very small particles of pure Ag were detected in Adyar sediments at station A6 using SEM-EDX observations (Figure 2). Their angular shape and small size (<8–10 μm) linked with their mono-elemental composition can be associated with very low sedimentary transport

and an anthropogenic origin. In [39], it was considered that no trace element signal trend should be attributed to a texture variation due to changes in sediment transportation and deposition throughout the entire stretch of both rivers.

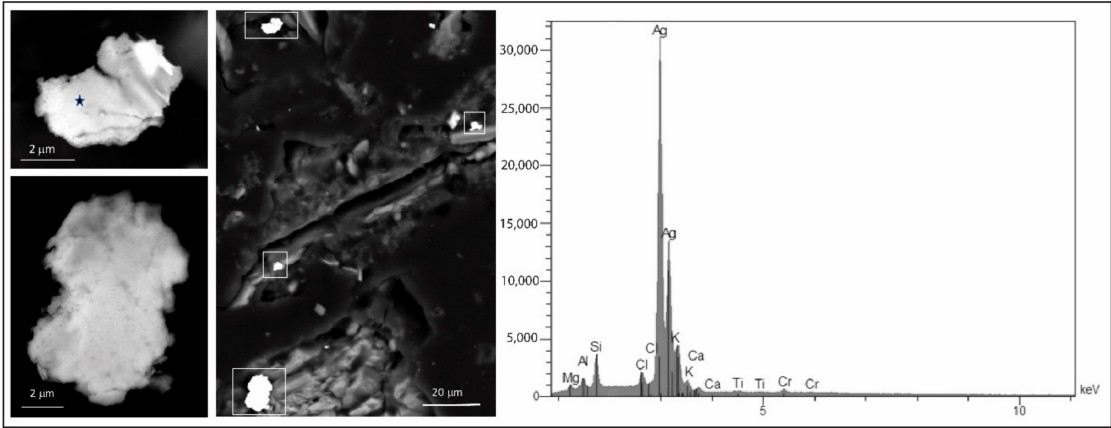

**Figure 2.** Particles of metallic Ag (white squares) in Adyar river sediment (scanning electron microscope observations) at the A6 station showing a mainly Ag composition (Energy Dispersive X-Ray Spectroscopy taken in the star location).

Hence, the increase in the mean $I_{\mathrm{geo}}$ values of Ag, Cd and Hg is clearly significant in the Chennai urban area for both rivers when compared with the upstream reaches (Figure 3a,b). These three trace elements can thus be used as urban pollution tracers in Chennai megacity.

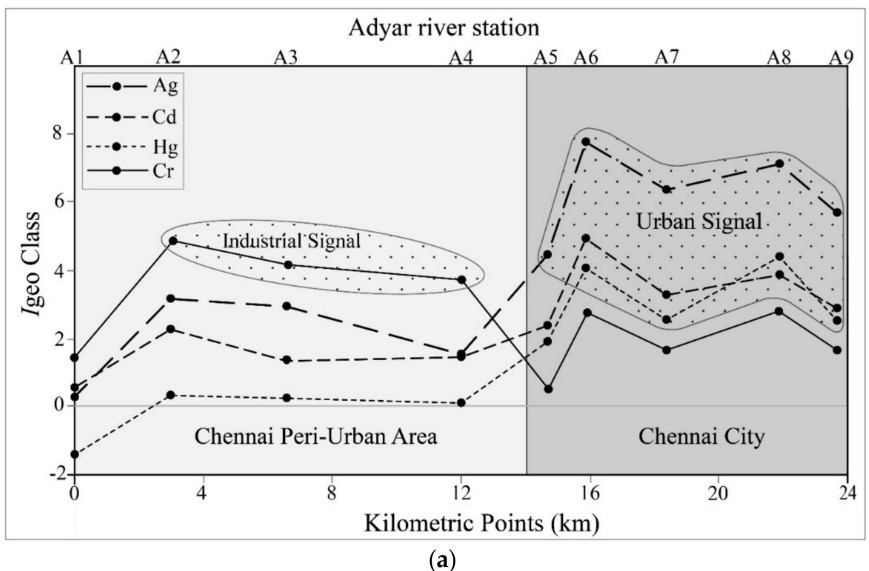

(a)

**Figure 3.** *Cont.*

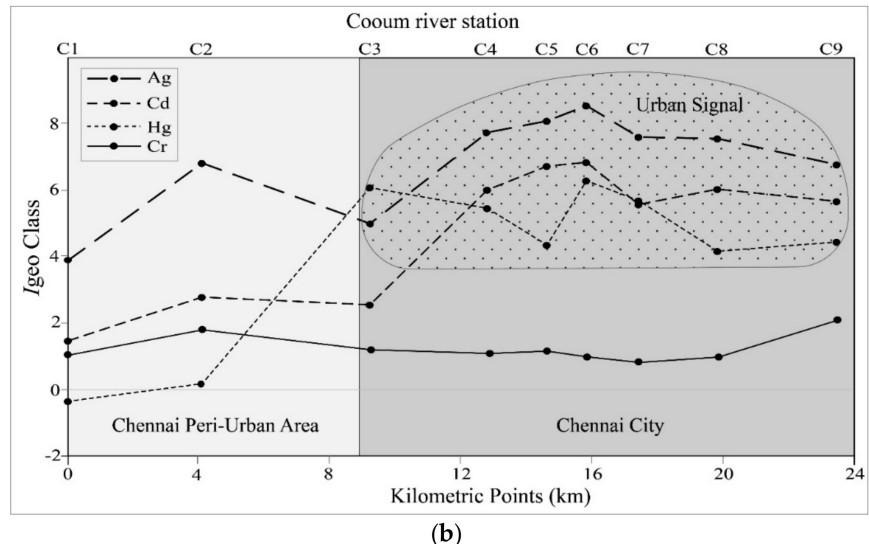

(**b**)

**Figure 3.** (**a**) Geoaccumulation Index ($I_{geo}$) value of trace elements (Ag, Cd, Cr and Hg) in Adyar river sediments; (**b**) Geoaccumulation Index ($I_{geo}$) value of trace elements (Ag, Cd, Cr and Hg) in Cooum river sediments.

### 3.3. Assessment of a Global Urban Footprint and Identification of Possible Socioeconomic Drivers

The Geochemical Urban Footprint Index (GUFI) is an index that was specifically developed for this study. This new index combines a spatial reference (here, the most upstream station is considered as less impacted) and the three main urban pollution tracers, Ag, Cd and Hg, selected based on the results of $I_{geo}$ analyses. The GUFI analysis was designed to represent a global urban geochemical footprint, independent from specific human activities or local effluents. The GUFI levels of contamination for each river station were then calculated (Table 3) and associated to population density and identified polluting hotspots at the ward scale (Figure 4).

**Table 3.** Geochemical Urban Footprint Index (GUFI) classification in the river sediment of Chennai City.

| GUFI Range | Level of Contamination | Adyar River | Cooum River | Urban Drain |
|---|---|---|---|---|
| 1–10 | Extremely Low Contamination | A1, A2, A3, A4 | C1, C2 | CSC |
| 10–25 | Low Contamination | A5 | C8, C9 | - |
| 25–50 | Moderate Contamination | A7, A9 | C3, C4, C5, C7 | RP, BC, GR |
| 50–75 | High Contamination | - | C6 | YMCA |
| 75–100 | Very High Contamination | A8 | - | - |
| >100 | Extremely High Contamination | A6 | - | NMR |

In the entire Chennai peri-urban area, the GUFI results show that contamination of the sediment is low in both rivers, except C8, C9 and A5, showing low contamination. The high values for C8 and C9 are unexplained, but in the case of A5, the most plausible explanation is the following: according to the official city limits, A5 station belongs to Chennai City but regarding its trace element concentration, this station could be geochemically related to the Chennai Metropolitan Development Authority. In the city of Chennai, the spatial trend of GUFI results indicates moderate to extreme contamination, particularly for downstream stations. For this reason, sediment samples taken from major drains in Chennai City were useful to identify sources of pollution and levels of contamination. Inside the city, all the sediment-sampled drains were contaminated from moderate to extreme levels. The results clearly show that pollution sources came directly from the urban area and were related to different anthropogenic activities, such as untreated sewage water outputs, unpunctual dumping areas, direct hospital sewage outlets and unplanned urban growth. The GUFI results of the drains studied in the

city reveal that contamination levels originated from various sources, in line with the overall trend of increasing population density, the city's development and punctual industrial hotspots.

Socioeconomic data were collected to identify the causes and origins of pollution hotspots. For administrative purposes, Chennai is divided into 155 wards, the limits of which are fixed by the Chennai Metropolitan Development Authority (CMDA) [90] (Figure 4). These polluting hotspots originate from industries, institutions, commercial buildings, sewage treatment plants, effluent treatment plants, etc., and information about them was obtained from the Tamil Nadu Pollution Control Board (TNPCB), which classifies hotspots into three categories, Red, Orange and Green, based on sewage/effluent discharge levels from various activities. Each category is subdivided into three classifications: large, medium and small. For this study, only large and medium Red hotspots were taken into account since they correspond to the most polluting hotspots with a huge amount of discharge of effluent. All the data on population and polluting hotspots attempt to frame the deprivation index and to identify the causes of Chennai's pollution. However, as one previous study [91] mentioned, Chennai is a complex city and a direct comparison between GUFI, population density and hotspots' spatial evolution is less valid than for other Indian megacities.

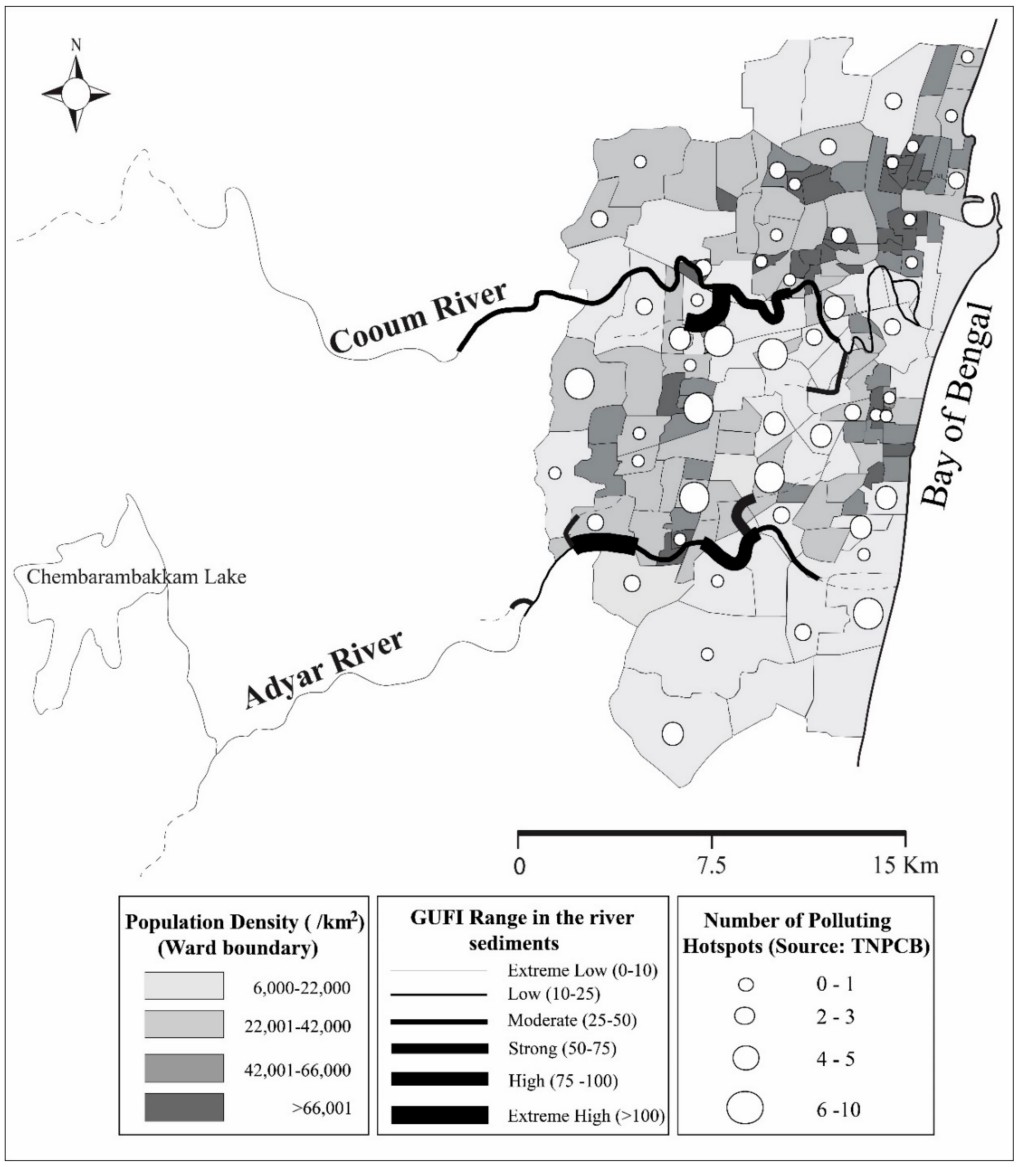

**Figure 4.** Longitudinal variation of the Geochemical Urban Footprint Index (GUFI) associated with population density and polluting hotspots at the ward scale in Chennai.

*3.4. Worldwide Comparative Analysis of Megacity Pollution Levels Using Sediment Quality Guidelines (SQGs)*

This comparative analysis was conducted on worldwide megacities and their respective geochemical state in order to assess trace element concentrations in the megacity of Chennai. The following megacities were statistically selected according to different criteria such as:

- Population range greater than 10 million
- The presence of an urban river flowing within the megacity limits
- Availability of data on trace element levels in the corresponding sediments

In the literature dedicated to urban sediments, concentrations of trace elements such as Cr, Cu, Ni, Pb and Zn are often surveyed in megacity rivers and canals as they are ubiquitous (Table 4). Information about population density and world population rank were also collected for all these megacities from [61], a robust and exhaustive database.

Concentrations below the TEC indicate no harmful effects and those above the PEC indicate that harmful effects can be expected [85]. In the selected megacity rivers (including the present study's rivers), the mean concentrations of Cu, Pb and Zn are lower than the proposed TEC levels (Table 4), indicating no harmful effects, in Jakarta (Ciliwung river, Indonesia) and Kinshasa (Congo river, DR Congo). Meanwhile, the mean concentrations of Cr, Cu, Pb, Ni and Zn in surface sediments for Tokyo-Yokohama (Tsurumi river, Japan), Jakarta (Ciliwung river, Indonesia) and Seoul (Han river, Korea) are higher than the PEC level. Other megacity rivers present concentrations lower than the PEC except for a few trace elements, including: (i) the Pearl river in Guangzhou (China) with Cu concentration exceeding the PEC level due to local sources from the urban area [19]; (ii) the Congo river in Kinshasa (DR Congo) with excessive Ni concentration due to human activities and the presence of uncontrolled landfills [92]; (iii) the Seine river in Paris (France) with excessive Pb concentration due to smelting and metallurgical industries [23]; (iv) the discharge canal in Lagos (Nigeria) with excessive Cr and Pb concentrations as a result of industrial and anthropogenic sources in the city [26]; (v) the Tham Luong canal in Ho Chi Minh City (Vietnam) with excessive Zn and Cr concentrations coming from chemical plant wastes released directly into the drain and indirectly from dry land [93]. Similarly, in South Asian countries, rivers in India and Bangladesh, such as the Yamuna river in Delhi, the Mithi river in Mumbai and the Buriganga river in Dhaka, have concentrations that are higher than the proposed PEC levels for Zn, Cr, Cu, Pb and Ni (except Pb in Delhi). These results are influenced by revealed factors such as population growth, industrial aggravation (automobile, tanneries, etc.), untreated industrial discharges and domestic waste water of households. All these controlled and uncontrolled discharges impact the biological life of the river [12,30,37].

All these selected megacities can be ranked according to (i) population rank, in which Tokyo-Yokohama (Japan) presents the largest population in the world (37.9 million), and (ii) megacity pollution rank. For this latter ranking, a comparative analysis of the selected megacities was carried out with the mean PEC quotients defined (Table 4), and the order from the most to the least impacted megacity is as follows: Jakarta > Kinshasa > Seoul > Tokyo–Yokohama > Lagos > Paris > Guangzhou > Chennai > Ho Chi Minh City > Delhi > Dhaka > Mumbai. This ranking of megacity pollution enables the cities to be divided into two groups:

- The first group refers to cities in Asia, the top six ranked cities: Mumbai (India), Dhaka (Bangladesh), Delhi (India), Ho Chi Minh City (Vietnam), Chennai (India), and Guangzhou (China). Similar levels of pollution can be observed in all of them due to rapid urban population growth and industrial growth. In terms of geochemical footprint, emerging cities that combine massive population growth and industrial activities have changed from traditional practices to modern economical activities. Chennai is the fifth most polluting megacity worldwide. It is on a par with the level of pollution of Ho Chi Minh City (Vietnam) and Guangzhou (China).
- The second group of megacities is composed of Paris (France), Lagos (Nigeria), Tokyo-Yokohama (Japan), Seoul (Korea) and Kinshasa (DR Congo). These five cities have similar pollution levels.

However, Nigeria and DR Congo are in developing countries, whereas France, Japan and Korea are developed. In upcoming decades, if such development in Nigeria and DR Congo continues, Lagos and Kinshasa may result in pollution levels of the first group. On the other hand, the Seine in Paris (France) showed greater pollution during the 1980s but with the introduction of strict regulations on water bodies' pollution, its status has greatly improved. Similarly, in Tokyo-Yokohama (Japan) and Seoul (Korea), pollution levels must obey strict regulations.

-　In relation to the data for Jakarta (Indonesia), the sediment concentration seems too low considering that this megacity presents the second highest population in the world and the water quality of the river is very poor due to discharging of municipal waste [94]. So, the study [95] which provided the result for Ciliwung river sediments appears suspicious as the Pb concent in the Ciliwang river is <1 ppm. Normally, the mean continental crust is minimum 20 times higher than the result for the Ciliwung river.

The comparison of ubiquitous trace element pollution of megacities is presented in Figure 5. The Chennai sediment quality (nine samples each from the Adyar and Cooum rivers) was compared to worldwide megacity river sediments (11 megacities) using PEC values (Figure 5). The three plots represent the Adyar river, Cooum river and the mean value for worldwide megacities' rivers, and median concentrations are compared with the PEC threshold. The median concentrations of Zn and Pb are below the PEC threshold in the three cases, and other trace elements (Cr and Ni) are also below the PEC level only in the Cooum river while the remaining rivers (Adyar and other megacities) are above the PEC levels. Higher PEC level concentrations are triggered by the influence of industrial activities. For example, in the Adyar river, the Cr concentration level is 1517 mg/kg at station A2 (Table 1), and this is an effect of tanneries from the Chromepet area (Figure 1).

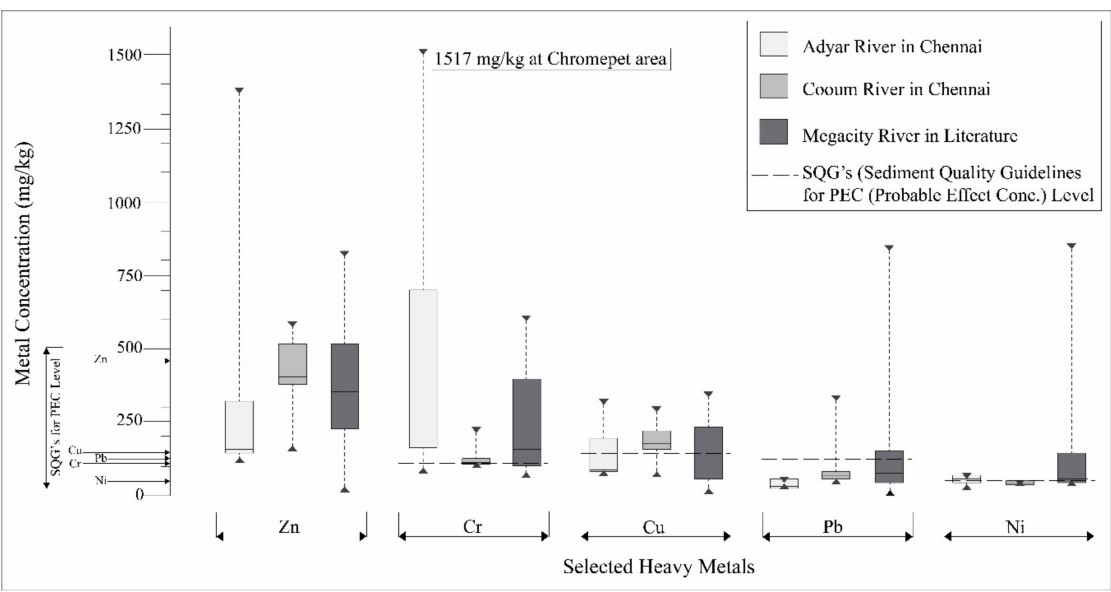

**Figure 5.** Box plots of selected ubiquitous trace elements comparing Chennai sediment quality to that of worldwide megacity sediments using SQGs.

**Table 4.** Mean concentrations of trace elements in surface sediments of various megacity rivers and canals and their associated megacity pollution ranking based on sediment quality guidelines (SQGs).

| Name of the Megacity | World Population Rank (April 2017) | Population (millions) | Population Density (/km²) | Name of the River | Trace Element (mg/kg) | | | | | References | Megacity Pollution Rank * |
|---|---|---|---|---|---|---|---|---|---|---|---|
| | | | | | Zn | Cr | Cu | Pb | Ni | | |
| SQGs for TEC level | | | | | 121 | 43 | 32 | 36 | 23 | [87] | |
| SQGs for PEC level | | | | | 459 | 111 | 149 | 128 | 49 | [87] | |
| Tokyo-Yokohama, Japan | 1 | 37.900 | 4400 | Tsurumi river | 381 | 103 | 133 | 41 | 37 | [27] | 9 |
| Jakarta, Indonesia | 2 | 31.760 | 9600 | Ciliwung river | 10 | - | 2.8 | 0.8 | - | [95] | 12 |
| Delhi, India | 3 | 26.495 | 12,000 | Yamuna river | 561 | 394 | 275 | 76 | 159 | [37] | 3 |
| Seoul, Korea | 5 | 24.105 | 8800 | Han river | 225 | 84 | 55 | 45 | 34 | [47] | 10 |
| Mumbai, India | 8 | 22.885 | 26,000 | Mithi river | - | 477 | - | 849 | 860 | [30] | 1 |
| Guangzhou, China | 13 | 19.075 | 5000 | Pearl river | 388 | 97 | 352 | 103 | - | [19] | 6 |
| Dhaka, Bangladesh | 15 | 16.820 | 45,700 | Buriganga river | 836 | 610 | 232 | 476 | 125 | [28] | 2 |
| Lagos, Nigeria | 24 | 13.360 | 9400 | Discharge canal | 319 | 157 | 68 | 130 | 48 | [26] | 8 |
| Kinshasa, DR Congo | 28 | 11.855 | 20,300 | Congo river | 50 | 59 | 24 | 9.3 | 58 | [92] | 11 |
| Paris, France | 31 | 10.950 | 3700 | Seine river | 231 | - | 56 | 169 | - | [23] | 7 |
| Ho Chi Minh City, Vietnam | 35 | 10.380 | 6600 | Tham Luong canal | 719 | 256 | - | 55 | - | [2] | 4 |
| Chennai, India | 36 | 10.265 | 9900 | Adyar river | 381 | 488 | 147 | 41 | 52 | Present Study | 5 |
| Chennai, India | 36 | 10.265 | 9900 | Cooum river | 409 | 131 | 180 | 104 | 43 | Present Study | 5 |

* see text for calculation of ranking based on the mean PEC quotient for each river.

Similarly, the worldwide megacity concentration of the element Cr is higher in the megacities of Delhi, Mumbai, Dhaka, Lagos, and Ho Chi Minh city, with multi-source impacts of industrial activities. The same impact was observed in worldwide megacities for Ni concentration. The median concentration of Cu is above the PEC level in the Cooum river and below the PEC level in the Adyar and worldwide megacities' rivers. In these megacities' comparison, high Cr and Ni concentrations are observed in many megacities whereas high Cu, Pb and Zn concentrations are observed in very few megacities and are commonly associated with urban pollution. The megacities have a trend pattern in the following order of ubiquitous trace elements: Zn < Cr < Cu < Pb < Ni. Most of these element concentrations originated from various industrial effluents and subsequently changed the geochemical features of the urban environment. This kind of worldwide megacity river sediment comparison, taking into account different pollution levels, is new in relation to previous research reports.

## 4. Conclusions

This study was conducted to determine and understand sediment quality in the two main rivers in Chennai megacity. The overall results of sediment concentrations for the trace elements Ag, As, Cd, Cr, Cu, Hg, Ni, Pb and Zn indicate an extremely polluted urban fluvial system. The Geoaccumulation Index ($I_{geo}$) revealed that the concentrations of Ag, Cd and Hg were present at levels indicating a highly to extremely polluted level for sediments in the urban area compared to other trace elements. The concentration of Cr, however, did indicate an extremely polluted state in the most upstream part of the Adyar river due to the influence of the tanning district. In general, this extremely polluted state can be linked to various pollutants discharged via diffuse urban and industrial point sources operating in and around the river bed. Thus, Ag, Cd and Hg concentrations can be used as urban tracers, and Cr concentration can be considered as an industrial marker in Chennai megacity.

The Geochemical Urban Footprint Index (GUFI), specifically developed in this study, was calculated for Chennai by combining the concentrations of these three key trace elements (Ag, Cd and Hg), compared to the most upstream remote station. The GUFI analysis was designed to represent a global urban geochemical footprint, regardless of specific human activities and local effluents, and it appears to be an indicator of anthropogenic impact in Chennai's rivers. The GUFI results show that the Chennai peri-urban area sediment has an extremely low level of contamination, indicating that these stations are not influenced by urban activity. Downstream in Chennai City, GUFI trends for both rivers and drains show moderately to extremely high contamination, particularly for sediment concentrations at stations A6 and A8 on the Adyar river, where levels are very high to extremely high. This increase in GUFI levels in the urban area is directly related to the overflow of combined sewers as shown by the analysis of highly impacted sediments in sampled urban drains. It can also be linked to an overall trend of population and industrial increases. However, this correlation cannot be direct as no upstream to downstream gradient could be evidenced with population density nor with the number of identified point pollution hotspots. Therefore, the new GUFI developed here can provide an alternative to the $I_{geo}$ to assess and quantify the pollution severity status of any megacity river that is influenced by contamination due to urbanization. The surface sediment concentrations of ubiquitous trace elements (Cr, Cu, Ni, Pb, and Zn) in various rivers and drains of megacities worldwide were compared regarding their pollution ranking (mean PEC quotient). According to this megacity pollution ranking, Chennai is ranked fifth, with the first six rankings all being megacities in Asian countries. This ranking sets Chennai on a par in terms of pollution with other highly populated megacities in the world, such as Ho Chi Minh City (Vietnam) and Guangzhou (China). Their pollution levels are similar in terms of reflecting the geochemical footprint of emerging cities combining massive population growth and industrial activity. Hence, the newly developed index, GUFI, is useful over the $I_{geo}$ to assess and quantify the pollution severity status of any megacity river influenced by emerging contamination due to urbanization. The pollution of Chennai's waterways has become both an eyesore and a serious source of disease in the city. The results of this study indicate that monitoring and immediate control measures must be taken to avoid further potential pollution of river sediment with toxic metals and

to ascertain the long-term effects on the environment. Further work should also be conducted to investigate the seasonal variability of toxic elements in these water bodies to mitigate the pollution of water resources in Chennai megacity.

**Author Contributions:** Conceptualization: S.P.S., M.D., C.G., A.N.P.K. and S.R.; Methodology: S.P.S. and M.D.; Software: S.P.S. and M.D.; Investigation: M.D., A.N.P.K. and C.G.; Resources: M.D.; Data: S.P.S., M.D., C.G., N.S. and A.N.P.K.; Writing—original draft preparation: S.P.S.; Writing—Review and editing: S.P.S., M.D., C.G., S.R. and A.N.P.K.; Supervision: M.D. and A.N.P.K.; Project administration: M.D. and C.G.; Funding acquisition: M.D. and C.G.

**Funding:** This work was supported by the ARCUS program funded by Regional Centre Val de Loire and the French Foreign Ministry, France.

**Acknowledgments:** The authors would like to thank the SARM-CPRG laboratory in Nancy (France) for conducting the geochemical analyses, SEM observations were obtained with the assistance of the IBiSA Electron Microscopy Facility of the University of Tours (Tours, France), and we would like to give a special thanks to all the support by members of the GeHCO Laboratory, University of Tours (Tours, France) and Centre for Water Resources, Anna University (Chennai, India) in providing the necessary facilities and technical assistance for this work.

**Conflicts of Interest:** The authors declare no conflict of interest.

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
