# Peer review of "Geochemical Footprint of Megacities on River Sediments: A Case Study of the Fourth Most Populous Area in India, Chennai"

_minerals, doi:10.3390/min9110688_

Round 1

Reviewer 1 Report

My comments and suggestions were followed almost to the letter.

However, there are still some misprints left.

For the details please see the attached file. The problematic parts are highlighted in yellow and additional comments were added.

Author Response

Dear reviewer,

Following the highlighted parts, we modified the text. See below :

Line 108 "the" has been removed and d replaced by D in word "Drainage"

Line 160 "using" is missing

Line 236 "the" has been removed and l replaced by L in word "Low"

Line 407 to 4011 the sentence is entirely rephrased as follow "In Jakarta, Pb concentration in Ciliwung river is < 1ppm. Normally, the mean continental crust is minimum 20 times higher than the result of Ciliwung river. Nevertheless, this megacity presents the second highest population in the world and water quality of the river is very poor due to discharging of municipal waste [94]. So, we are astonished by the results presented in Yasuda et al. [95] and we suspect a problem of representativeness of the collected bed sediment"

Line 436 to 437 : the sentence is removed. 

Reviewer 2 Report

The paper is worthy of publication as it reports important database on contaminated sediments of two fluvial systems close to Chennai city. The manuscript quality can be improved if the following concerns are addressed:

I would like a hypothesis stated upfront, which is testable. The inclusions of long. and lat. in Figs. 1  and 4 would be useful guide to the study area for those not familiar with the Indian geography.  While the analysis of the metals in the <62um size of sediments is acceptable (to normalize effects of inter-sample grain size differences), it would have been more appropriate to separate the above size by wet sieving of gross sediment through a nylon screen rather than pulverizing dry sample and sieving, which could disaggregate sand.  Statistical analyses on the database are invariably missing. Lack of these tests renders some of the conclusions qualitative. For example, are the concentrations of metals in the upper and lower fluvial reaches (arbitrarily defined) really different? Cluster analysis/SMDA and surface trend analysis of the metal data would clarify if there are station groupings. How are high metal levels in some upstream stations explained? Name the Reference Standards used. A brief discussion would be appropriate on the potential adverse impacts of the contaminated sediments on the region's fluvial benthic and demersal species. Refer to Long et al. (1998, Environ. Toxicol. Chem., 17:714-727), USEPA guideline, among many others.

Author Response

Dear reviewer,

See below the response to the points you raised in your review :

In figure 1 and 4, we will find a small map inserted inside the map focusing on the Chennai area. It is very helpful for those who need to locate at a global scale. 

Wet sieving is a very interesting technic we could investigate into our lab but we were facing to a big issue : the distilled water can desorb metals from solids particles. This is the reason why we privileged methods on dry sediments

We asked ourselves the question of the statistical analyzes and we worked with specialists of this type of analysis who recommended us not to do it. Indeed, the number of samples is not sufficient to apply robust methods of statistics. Nevertheless, our data can be analyzed in term of general to local trends and pollution trajectories.

Data regarding organic compounds are more abundant so, we will do some statical analysis in the near future, in the forthcoming article dealing with organic compounds. 

Unexpected high metal concentration in some upstream station remain unclear even for the authors. This is an hypothesis : hydrosedimentary functioning of the rivers brings on a grain size gradient from upstream to downstream. So, the amount of fine particles in bed sediments collected upstream in less than the amount of fine particles collected downstream, due to this natural process. Sieving, calibration and normalization could avoid this grain size effect but it could be part of the explanation.  

We will try to discuss potential adverse impacts of the contaminated sediments in the near future, into the forthcoming article, as we will include organic compounds. 

This manuscript is a resubmission of an earlier submission. The following is a list of the peer review reports and author responses from that submission.

Round 1

Reviewer 1 Report

Review of the manuscript: Geochemical Footprint of Megacities on River Sediments: A Case Study of the Fourth Most Populous Area in India, Chennai

The manuscript is dealing with trace element pollution in Megacity area and the ranking of the city according to pollution level.

The paper is well written and rather easy to follow, with some minor exceptions. I would suggest to the authors to check the English since some of the sentences were not easy to follow. You can find some suggestions in the list bellow.

Minor remarks

Pg 1, line 23: Use capital letter for silver

Pg 3, line 10: Use capital letter for the

Pg 3, lines 121-122: The sentence „The upstream portions of two rivers are dry outside the non-monsoon period“ is not clear. I would expect them to be dry outside the monsoon period and not outside the non-monsoon period. Please clarify.

Pg 3, lines 125-126: „As river slopes are very low, sediment transportation is also reduced and dilution by various water nor particle inputs is not very effective.“ Again a rather awkward sentence. Please rephrase.

Pg 4., line 140 and after: You stated that flood deposits were sampled under water at a depth 0-15 cm.  I would expect flood deposits to be spread on the flooplain, i.e. to be dry. Besides, the following sentence is not clear 'This upper layer of surface sediment is considered as recent deposition as one hydrological cycle/one-year and flood deposits.' Please clarify.

Pg 5, line 161: Remove dot after the word 'using'

Pg 5, line 166: 't' is missing in geoaccumulation

Pg 7, line 237: Reverse the word order - On the opposite, the minimum concentrations of As, Cr and Ni were measured in more downstream reaches, especially at the mouth of the rivers.

Pg 7, line 243: Use plural - '...their concentrations maxima...'

Pg 7, line 264-265:  I believe your sentence should read '...more elevated concentrations than river sediments....'

Pg 8, line 265: Reverse word order – '...river sediments..'

Pg 13, line 369: Close parenthesis after 'France'

Pg 13, the paragraph starting with line 402: The paragraph is unclear. You state that the data for Jakarta are out of range (too low) and at the same time that ' This megacity presents the second highest population in the world and water quality of the river is very poor due to discharging municipal waste [67] '. It's a contradiction. Please clarify.

The following sentence is particularly confusing ' So, this study [98] will not be taken into account for Ciliwung river sediments.' Which study will not be taken into account? The study of  Palupi et al (ref 67) or the study of Yasuda et al (98)?

Pg 14, Table 4. The names of the rivers should be written in capital letters (Yamuna, Ciliwung)

Pg 15, line 424: The sentence should be 'The same impact was observed in worldwide megacities for Ni concentration.'

Pg 15, line 431: I'm not quite sure what was ment here. I suggest to shorten the sentence and make  it  more understandable. The way it is written now is not very clear.

Pg 16, line 462: It shoud read  ...ubiquitous trace elements...

Reviewer 2 Report

I have read carefully the manuscript (Minerals-538088) which deals with an interesting environmental issue regarding the contamination of river sediments. Although the manuscript is well written and fit the Aims and Scope of the journal there are several issues that need clarification. My main concerns result from the development of a new urban index specifically for the exact study area. Moreover, the use of TrElems like Ag, Cd and Hg in the newly imported index, is rather an incorrect approach as their high contents are due to local sources and they do not exhibit constant high contents in the entire urban area (see stations A9 & C9). Furthermore, the authors avoid identifying possible sources for elevated TrElems contents. On the other hand, the authors totally ignore the local geology. Thought they mention the absence of local geochemical data, they could use the upstream (less impacted) samples as local background values for the calculation of pollution indices. Moreover, the use of mean PEC quotient for direct comparison among urban cities is incorrect. The direct comparison is difficult due to differences in analytical methods, sampling protocols etc. Further, a statistical analysis regarding the significance of the differences among rivers or upstream-downstream samples is also missing. Finally, the references must be numbered in order of appearance in the text. Therefore, the manuscript is not suitable for publication in its present form and requires major revision. Detailed are given below.

P4, L150-151: explain why the <63μm size fraction was used. What do you mean to limit grain size effect?

P5, L178-180: Calculate Igeo based on less contaminated sample.

P7, L235-237: What about station C2 which present elevated Ag, Pb and even Zn contents?

P7, L237-238: Not true for Cooum river (see station C9).

P7, L240-242: This is not true. For example if we focus on samples A9 & C9 they exhibit notably lower contents. The samples are not presenting a constant decrease downwards. Instead some samples probably due to local sources are more impacted.  

P8, L248-249: Decreased compared to what? All studied TrElems (except Ni) exhibit increased concentrations in C9.

P10, L315-318: GUFI index independent upon specific human activities and local effluents. But it is evident that the elevated TrElems contents in specific downstream stations are due to local sources and specific anthropogenic activities.

P11, L322-323: Samples C8, C9 & A5 show low contamination

P13, L382-384: from least to most impacted. However, as I mentioned earlier, a direct comparison is quite difficult.

P15, L409: Figure 5 just illustrate the data from Table 4. Remove.

Table 2: A box-plot diagram would be more informative exhibiting the range of Igeo values.

Figure 2: Why the authors have excluded the rest of the TrElems. What about Cu and Zn in Cooum river?